# IMPLICIT CAUSAL MODELS FOR GENOME-WIDE ASSOCIATION STUDIES

## ABSTRACT

Progress in probabilistic generative models has accelerated, developing richer models with neural architectures, implicit densities, and with scalable algorithms for their Bayesian inference. However, there has been limited progress in models that capture causal relationships, for example, how individual genetic factors cause major human diseases. In this work, we focus on two challenges in particular: How do we build richer causal models, which can capture highly nonlinear relationships and interactions between multiple causes? How do we adjust for latent confounders, which are variables influencing both cause and effect and which prevent learning of causal relationships? To address these challenges, we synthesize ideas from causality and modern probabilistic modeling. For the first, we describe *implicit causal models*, a class of causal models that leverages neural architectures with an implicit density. For the second, we describe an implicit causal model that adjusts for confounders by sharing strength across examples. In experiments, we scale Bayesian inference on up to a billion genetic measurements. We achieve state of the art accuracy for identifying causal factors: we significantly outperform existing genetics methods by an absolute difference of 15-45.3%.

## 1 INTRODUCTION

Probabilistic models provide a language for specifying rich and flexible generative processes (Pearl, 1988; Murphy, 2012). Recent advances expand this language with neural architectures, implicit densities, and with scalable algorithms for their Bayesian inference (Rezende et al., 2014; Tran et al., 2017). However, there has been limited progress in models that capture high-dimensional causal relationships (Pearl, 2000; Spirtes et al., 1993; Imbens & Rubin, 2015). Unlike models which learn statistical relationships, causal models let us manipulate the generative process and make counterfactual statements, that is, what would have happened if the distributions changed.

As the running example in this work, consider genome-wide association studies (GWAS) (Yu et al., 2005; Price et al., 2006; Kang et al., 2010). The goal of GWAS is to understand how genetic factors, i.e., single nucleotide polymorphisms (SNPs), cause traits to appear in individuals. Understanding this causation both lets us predict whether an individual has a genetic predisposition to a disease and also understand how to cure the disease by targeting the individual SNPs that cause it.

With this example in mind, we focus on two challenges to combining modern probabilistic models and causality. The first is to develop richer, more expressive causal models. Probabilistic causal models represent variables as deterministic functions of noise and other variables, and existing work usually focuses on additive noise models (Hoyer et al., 2009) such as linear mixed models (Kang et al., 2010). These models apply simple nonlinearities such as polynomials, hand-engineered low order interactions between inputs, and assume additive interaction with, e.g., Gaussian noise. In GWAS, strong evidence suggests that susceptibility to common diseases is influenced by epistasis (the interaction between multiple genes) (Culverhouse et al., 2002; McKinney et al., 2006). We would like to capture and discover such interactions. This requires models with nonlinear, learnable interactions among the inputs and the noise.

The second challenge is how to address latent population-based confounders. In GWAS, both latent population structure, i.e., subgroups in the population with ancestry differences, and relatedness among sample individuals produce spurious correlations among SNPs to the trait of interest. Existing methods correct for this correlation in two stages (Yu et al., 2005; Price et al., 2006; Kang et al.,

2010): first, estimate the confounder given data; then, run standard causal inferences given the estimated confounder. These methods are effective in some settings, but they are difficult to understand as principled causal models, and they cannot easily accommodate complex latent structure.

To address these challenges, we synthesize ideas from causality and modern probabilistic modeling. For the first challenge, we develop *implicit causal models*, a class of causal models that leverages neural architectures with an implicit density. With GWAS, implicit causal models generalize previous methods to capture important nonlinearities, such as gene-gene and gene-population interaction. Building on this, for the second challenge, we describe an implicit causal model that adjusts for population-confounders by sharing strength across examples (genes). We derive conditions that prove the model consistently estimates the causal relationship. This theoretically justifies existing methods and generalizes them to more complex latent variable models of the confounder.

In experiments, we scale Bayesian inference on implicit causal models on up to a billion genetic measurements. Validating these results are not possible for observational data (Pearl, 2000), so we first perform an extensive simulation study of 11 configurations of 100,000 SNPs and 940 to 5,000 individuals. We achieve state of the art accuracy for identifying causal factors: we significantly outperform existing genetics methods by an absolute difference of 15-45.3%. In a real-world GWAS, we also show our model discovers real causal relationships—identifying similar SNPs as previous state of the art—while being more principled as a causal model.

## 1.1 RELATED WORK

There has been growing work on richer causal models. Louizos et al. (2017) develop variational auto-encoders for causality and address local confounders via proxy variables. Our work is complementary: we develop implicit models for causality and address global confounders by sharing strength across examples. In other work, Mooij et al. (2010) propose a Gaussian process over causal mechanisms, and Zhang & Hyvärinen (2009) study post-nonlinear models, which apply a nonlinearity after adding noise. These models typically focus on the task of causal discovery, and they assume fixed nonlinearities (post-nonlinear models) or impose strong smoothness assumptions (Gaussian processes) which we relax using neural networks. In the potential outcomes literature, much recent work has considered decision trees and neural networks (e.g., Hill (2011); Wager & Athey (2015); Johansson et al. (2016)). These methods tackle a related but different problem of balancing covariates across treatments.

Causality with population-confounders has primarily been studied for genome-wide association studies (GWAS). A popular approach is to first, estimate the confounder using the top principal components of the genotype matrix of individuals-by-SNPs; then, linearly regress the trait of interest onto the genetic factors and these components (Price et al., 2006; Astle & Balding, 2009). Another approach is to first, estimate the confounder via a "kinship matrix" on the genotypes; then, fit a linear mixed model of the trait given genetic factors, and where the covariance of random effects is the kinship matrix (Yu et al., 2005; Kang et al., 2010). Other work adjusts for the confounder via admixture models and factor analysis (Song et al., 2015; Hao et al., 2016). This paper builds on all these methods, providing a theoretical understanding about when causal inferences can succeed while adjusting for latent confounders. We also develop a new causal model with nonlinear, learnable gene-gene and gene-population interactions; and we describe a Bayesian inference algorithm that justifies the two-stage estimation.

The problem of epistasis, that is, nonadditive interactions between multiple genes, dates back to classical work on epigenetics by Bateson and R.A. Fisher (Fisher, 1918). Primary methods for capturing epistasis include adding interactions within a linear model, permutation tests, exhaustive search, and multifactor dimensionality reduction (Cordell, 2009). These methods require hand-engineering over all possible interactions, which grows exponentially in the number of genetic factors. Neural networks have been applied to address epistasis for epigenomic data, such as to predict sequence specificities of protein bindings given DNA sequences (Alipanahi et al., 2015). These methods use discriminative neural networks (unlike neural networks within a generative model), and they focus on prediction rather than causality.

## 2 IMPLICIT CAUSAL MODELS

We describe the framework of probabilistic causal models. We then describe implicit causal models, an extension of implicit models for encoding complex, nonlinear causal relationships.

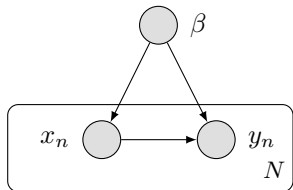 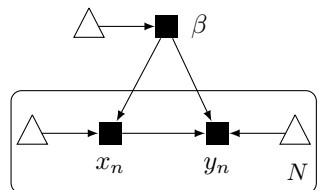

**Figure 1:** Probabilistic causal model. **(left)** Variable $x$ causes $y$ coupled with a shared variable $\beta$. **(right)** A more explicit diagram where variables (denoted with a square) are a deterministic function of other variables and noise $\epsilon$ (denoted with a triangle).

## 2.1 PROBABILISTIC CAUSAL MODELS

Probabilistic causal models (Pearl, 2000), or structural equation models, represent variables as deterministic functions of noise and other variables. As illustration, consider the causal diagram in Figure 1. It represents a causal model where there is a global variable

$$\beta = f_\beta(\epsilon_\beta), \qquad \epsilon_\beta \sim s(\cdot),$$

and for each data point $n = 1, \ldots, N$,

$$
\begin{aligned}
x_n &= f_x(\epsilon_{x,n}, \beta), & \epsilon_{x,n} &\sim s(\cdot) \\
y_n &= f_y(\epsilon_{y,n}, x_n, \beta), & \epsilon_{y,n} &\sim s(\cdot).
\end{aligned}
\tag{1}
$$

The noise $\epsilon$ are background variables, representing unknown external quantities which are jointly independent. Each variable $\beta, x, y$ is a function of other variables and its background variable.

We are interested in estimating the causal mechanism $f_y$. It lets us calculate quantities such as the causal effect $p(y \mid \mathrm{do}(X = x), \beta)$, the probability of an outcome $y$ given that we force $X$ to a specific value $x$ and under fixed global structure $\beta$. This quantity differs from the conditional $p(y \mid x, \beta)$. The conditional takes the model and filters to the subpopulation where $X = x$; in general, the processes which set $X$ to that value may also have influenced $Y$. Thus the conditional is not the same as if we had manipulated $X$ directly (Pearl, 2000).

Under the causal graph of Figure 1, the adjustment formula says that $p(y \mid \mathrm{do}(x), \beta) = p(y \mid x, \beta)$. This means we can estimate $f_y$ from observational data $\{(x_n, y_n)\}$, assuming we observe the global structure $\beta$. For example, an additive noise model (Hoyer et al., 2009) posits

$$y_n = f(x_n, \beta \mid \theta) + \epsilon_n, \qquad \epsilon \sim s(\cdot),$$

where $f(\cdot)$ might be a linear function of the concatenated inputs, $f(\cdot) = [x_n, \beta]^\top \theta$, or it might use spline functions for nonlinearities. If $s(\cdot)$ is standard normal, the induced density for $y$ is normal with unit variance. Placing a prior over parameters $p(\theta)$, Bayesian inference yields

$$p(\theta \mid \mathbf{x}, \mathbf{y}, \beta) \propto p(\theta) p(\mathbf{y} \mid \mathbf{x}, \theta, \beta).
\tag{2}$$

The right hand side is a joint density whose individual components can be calculated. We can use standard algorithms, such as variational inference or MCMC (Murphy, 2012).

A limitation in additive noise models that they typically apply simple nonlinearities such as polynomials, hand-engineered low-order interactions between inputs, and assume additive interaction with, e.g., Gaussian noise. Next we describe how to build richer causal models which relax these restrictions. (An additional problem is that we typically don't observe $\beta$; we address this in § 3.)

## 2.2 IMPLICIT CAUSAL MODELS

Implicit models capture an unknown distribution by hypothesizing about its generative process (Diggle & Gratton, 1984; Tran et al., 2017). For a distribution $p(x)$ of observations $x$, recent advances define a function $g$ that takes in noise $\epsilon \sim s(\cdot)$ and outputs $x$ given parameters $\theta$,

$$x = g(\epsilon \mid \theta), \quad \epsilon \sim s(\cdot).
\tag{3}$$

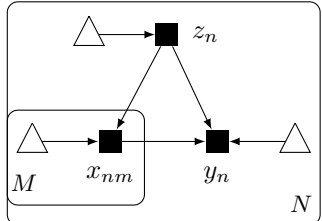 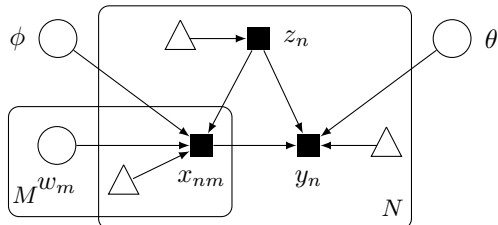

**Figure 2:** **(left)** Causal graph for GWAS. The population structure of SNPs for each individual ($z_n$) confounds inference of how each SNP ($x_{nm}$) causes a trait of interest ($y_n$). **(right)** Implicit causal model for GWAS (described in § 3.2). Its structure is the same as the causal graph but also places priors over parameters $\phi$ and $\theta$ and with a latent variable $w_m$ per SNP.

Unlike models which assume additive noise, setting $g$ to be a neural network enables multilayer, nonlinear interactions. Implicit models also separate randomness from the transformation; this imitates the structural invariance of causal models (Equation 1).

To enforce causality, we define an *implicit causal model* as a probabilistic causal model where the functions $g$ form structural equations, that is, causal relations among variables. Implicit causal models extend implicit models in the same way that causal networks extend Bayesian networks (Pearl & Verma, 1991) and path analysis extends regression analysis (Wright, 1921). They are nonparametric structural equation models where the functional forms are themselves learned.

A natural question is the representational capacity of implicit causal models. In general, they are universal approximators: we can use a fully connected network with a sufficiently large number of hidden units to approximate each causal mechanism. We describe this formally.

**Theorem** (Universal Approximation Theorem). *Let the tuple $(\mathcal{E}, V, F, s(\mathcal{E}))$ denote a probabilistic causal model, where $\mathcal{E}$ represent the set of background variables with probability $s(\mathcal{E})$, $V$ the set of endogenous variables, and $F$ the causal mechanisms. Assume each causal mechanism is a continuous function on the $m$-dimensional unit cube $f \in \mathcal{C}([0,1]^m)$. Let $\sigma$ be a nonconstant, bounded, and monotonically-increasing continuous function.*

*For each causal mechanism $f$ and any error $\delta > 0$, there exist parameters $\theta = \{\mathbf{v}, \mathbf{w}, \mathbf{b}\}$, for real constants $v_i, b_i \in \mathbb{R}$ and real vectors $w_i \in \mathbb{R}^m$ for $i = 1, \ldots, H$ and fixed $H$, such that the following function approximates $f$:*

$$g(x \,|\, \theta) = \sum_{i=1}^{H} v_i \sigma \left( w_i^T x + b_i \right), \qquad |g(x \,|\, \theta) - f(x)| < \delta \quad \text{for all } x \in [0,1]^m.$$

*The implicit model defined by the collection of functions $g$ and same noise distributions universally approximates the true causal model.*

(This directly follows from the approximator theorem of, e.g., Cybenko (1989).) A key aspect is that implicit causal models are not only universal approximators, but that we can use fast algorithms for their Bayesian inference (to calculate Equation 2). In particular, variational methods both scale to massive data and provide accurate posterior approximations (§ 4). This lets us obtain good performance in practice with finite-sized neural networks; § 5 describes such experiments.

## 3 IMPLICIT CAUSAL MODELS WITH LATENT CONFOUNDERS

We described implicit causal models, a rich class of models that can capture arbitrary causal relations. For simplicity, we assumed that the global structure is observed; this enables standard inference methods. We now consider the typical setting when it is unobserved.

### 3.1 CAUSAL INFERENCE WITH A LATENT CONFOUNDER

Consider the running example of genome-wide association studies (GWAS) (Figure 2). There are $N$ data points (individuals). Each data point consists of an input vector of length $M$ (measured SNPs), $x_n = [x_{n1}, \ldots, x_{nM}]$ and a scalar outcome $y_n$ (the trait of interest). Typically, the number of

measured SNPs $M$ ranges from 100,000 to 1 million and the number of individuals $N$ ranges from 500 to 10,000.

We are interested in how changes to each SNP $X_m$ cause changes to the trait $Y$. Formally, this is the causal effect $p(y \mid \mathrm{do}(x_m), x_{-m})$, which is the probability of an outcome $y$ given that we force SNP $X_m = x_m$ and consider fixed remaining SNPs $x_{-m}$. Standard inference methods are confounded by the unobserved population structure of SNPs for each individual, as well as the individual's cryptic relatedness to other samples in the data set. This confounder is represented as a latent variable $z_n$, which influences $x_{nm}$ and $y_n$ for each data index $n$; see Figure 2. Because we do not observe the $z_n$'s, the causal effect $p(y \mid \mathrm{do}(x_m), x_{-m})$ is unidentifiable (Spirtes et al., 1993).

Building on previous GWAS methods (Price et al., 2006; Yu et al., 2005; Astle & Balding, 2009), we build a model that jointly captures $z_n$'s and the mechanisms for $X_m \to Y$. Consider the implicit causal model where for each data point $n = 1, \dots, N$ and for each SNP $m = 1, \dots, M$,

$$
\begin{aligned}
z_n &= g_z(\epsilon_{z_n}), & \epsilon_{z_n} &\sim s(\cdot), \\
x_{nm} &= g_{x_m}(\epsilon_{x_{nm}}, z_n \mid w_m), & \epsilon_{x_{nm}} &\sim s(\cdot), \\
y_n &= g_y(\epsilon_{y_n}, x_{n,1:M}, z_n \mid \theta), & \epsilon_{y_n} &\sim s(\cdot).
\end{aligned}
\tag{4}
$$

The function $g_z(\cdot)$ for the confounder is fixed. Each function $g_{x_m}(\cdot \mid w_m)$ per SNP depends on the confounder and has parameters $w_m$. The function $g_y(\cdot \mid \theta)$ for the trait depends on the confounder and all SNPs, and it has parameters $\theta$. We place priors over the parameters $p(w_m)$ and $p(\theta)$.

Figure 2 (right) visualizes the model. It is a model over the full causal graph (Figure 2 (left)) and differs from the unconfounded case: Equation 2 only requires a model from $X \to Y$, and the rest of the graph is "ignorable" (Imbens & Rubin, 2015).

To estimate the mechanism $f_y$, we calculate the posterior of the outcome parameters $\theta$,

$$
p(\theta \mid \mathbf{x}, \mathbf{y}) = \int p(\mathbf{z} \mid \mathbf{x}, \mathbf{y}) p(\theta \mid \mathbf{x}, \mathbf{y}, \mathbf{z}) \, \mathrm{d}\mathbf{z}.
\tag{5}
$$

Note how this accounts for the unobserved confounders: it assumes that $p(\mathbf{z} \mid \mathbf{x}, \mathbf{y})$ accurately reflects the latent structure. In doing so, we perform inferences for $p(\theta \mid \mathbf{x}, \mathbf{y}, \mathbf{z})$, averaged over posterior samples from $p(\mathbf{z} \mid \mathbf{x}, \mathbf{y})$.

In general, causal inference with latent confounders can be dangerous: it uses the data twice (once to estimate the confounder; another to estimate the mechanism), and thus it may bias our estimates of each arrow $X_m \to Y$. Why is this justified? We answer this below.

**Proposition 1.** *Assume the causal graph of Figure 2 (left) is correct and that the true distribution resides in some configuration of the parameters of the causal model (Figure 2 (right)). Then the posterior $p(\theta \mid \mathbf{x}, \mathbf{y})$ provides a consistent estimator of the causal mechanism $f_y$.*

(See Appendix A for the proof.) Proposition 1 rigorizes previous methods in the framework of probabilistic causal models. The intuition is that as more SNPs arrive ("$M \to \infty$, $N$ fixed"), the posterior concentrates at the true confounders $z_n$, and thus we can estimate the causal mechanism given each data point's confounder $z_n$. As more data points arrive ("$N \to \infty$, $M$ fixed"), we can estimate the causal mechanism given any confounder $z_n$ as there are infinity of them.

**Connecting to Two-Stage Estimation.** Existing GWAS methods adjust for latent population structure using two stages (Astle & Balding, 2009): first, estimate the confounders $z_{1:N}$; second, infer the outcome parameters $\theta$ given the data set and the estimate of the confounders. To incorporate uncertainty, a Bayesian version would not use a point estimate of $z_{1:N}$ but the full posterior $p(z_{1:N} \mid \mathbf{x}, \mathbf{y})$; then it would infer $\theta$ given posterior samples of $z_{1:N}$. Following Equation 5, this is the same as joint posterior inference. Thus the two stage approach is justified as a Bayesian approximation that uses a point estimate of the posterior.

### 3.2 IMPLICIT CAUSAL MODEL WITH A LATENT CONFOUNDER

Above, we outlined how to specify an implicit causal model for GWAS. We now specify in detail the functions and priors for the confounders $z_n$, the SNPs $x_{nm}$, and the traits $y_n$ (Equation 4). Figure 2 (right) visualizes the model we describe below. Appendix B provides an example implementation in the Edward probabilistic programming language (Tran et al., 2016).

**Generative Process of Confounders** $z_n$**.** We use standard normal noise and set the confounder function $g_z(\cdot)$ to the identity. This implies the distribution of confounders $p(z_n)$ is standard normal. Their dimension $z_n \in \mathbb{R}^K$ is a hyperparameter. The dimension $K$ should be set to the highest value such that the latent space most closely approximates the true population structure but smaller than the total number of SNPs to avoid overfitting.

**Generative Process of SNPs** $x_{nm}$**.** Designing nonlinear processes that return matrices is an ongoing research direction (e.g., Lawrence (2005); Lloyd et al. (2012)). To design one for GWAS (the SNP matrix), we build on an implicit modeling formulation of factor analysis; it has been successful in GWAS applications (Price et al., 2006; Song et al., 2015). Let each SNP be encoded as a 0, 1, or 2 to denote the three possible genotypes. This is unphased data, where 0 indicates two major alleles; 1 indicates one major and one minor allele; and 2 indicates two minor alleles. Set

$$\text{logit } \pi_{nm} = z_n^\top w_m, \qquad x_{nm} = \mathbb{I}[\epsilon_1 > \pi_{nm}] + \mathbb{I}[\epsilon_2 > \pi_{nm}], \qquad \epsilon_1, \epsilon_2 \sim \text{Uniform}(0, 1).$$

This defines a $\text{Binomial}(2, \pi_{nm})$ distribution on $x_{nm}$. Analogous to generalized linear models, the Binomial's logit probability is linear with respect to $z_n$. We then sum up two Bernoulli trials: they are represented as indicator functions of whether a uniform sample is greater than the probability. (The uniform noises are newly drawn for each index $n$ and $m$.)

Assuming a standard normal prior on the variables $w_m$, this generative process is equivalent to logistic factor analysis. The variables $w_m$ act as "principal components," embedding the $M$-many SNPs within a subspace of lower dimension $K$.

Logistic factor analysis makes strong assumptions: linear dependence on the confounder and that one parameter per dimension has sufficient representational capacity. We relax these assumptions using a neural network over concatenated inputs,

$$\text{logit } \pi_{nm} = \text{NN}([z_n, w_m] \,|\, \phi).$$

Similar to the above, the variables $w_m$ serve as principal components. The neural network takes an input of dimension $2K$ and outputs a scalar real value; its weights and biases $\phi$ are shared across SNPs $m$ and individuals $n$. This enables learning of nonlinear interactions between $z_n$ and $w_m$, preserves the model's conditional independence assumptions, and avoids the complexity of a neural net that outputs the full $N \times M$ matrix. We place a standard normal prior over $\phi$.

**Generative Process of Traits** $y_n$**.** To specify the traits, we build on an implicit modeling formulation of linear regression. It is the mainstay tool in GWAS applications (Price et al., 2006; Song et al., 2015). Formally, for real-valued $y \in \mathbb{R}$, we model each observed trait as

$$y_n = [x_{n,1:M}, z_n]^\top \theta + \epsilon_n, \qquad \epsilon_n \sim \text{Normal}(0, 1),$$

This process assumes linear dependence on SNPs, no gene-gene and gene-population interaction, and additive noise. We generalize this model using a neural network over the same inputs,

$$y_n = \text{NN}([x_{n,1:M}, z_n, \epsilon] \,|\, \theta), \qquad \epsilon_n \sim \text{Normal}(0, 1).$$

The neural net takes an input of dimension $M + K + 1$ and outputs a scalar real value; for categorical outcomes, the output is discretized over equally spaced cutpoints. We also place a group Lasso prior on weights connecting a SNP to a hidden layer. This encourages sparse inputs: we suspect few SNPs affect the trait (Yuan & Lin, 2006). We use standard normal for other weights and biases.

## 4 LIKELIHOOD-FREE VARIATIONAL INFERENCE

We described a rich causal model for how SNPs cause traits and that can adjust for latent population-confounders. Given GWAS data, we aim to infer the posterior of outcome parameters $\theta$ (Equation 5). Calculating this posterior reduces to calculating the joint posterior of confounders $z_n$, SNP parameters $w_m$ and $\phi$, and trait parameters $\theta$,

$$p(z_{1:N}, w_{1:M}, \phi, \theta \,|\, \mathbf{x}, \mathbf{y}) \propto p(\phi)p(\theta) \prod_{n=1}^{N} \left[ p(z_n)p(y_n \,|\, x_{n,1:M}, z_n, \theta) \prod_{m=1}^{M} p(w_m)p(x_{nm} \,|\, z_n, w_m, \phi) \right].$$

This means we can use typical inference algorithms on the joint posterior. We then collapse variables to obtain the marginal posterior of $\theta$. (For Monte Carlo methods, we drop the auxiliary samples; for variational methods, it is given if the variational family follows the posterior's factorization.)

One difficulty is that with implicit models, evaluating the density is intractable: it requires integrating over a nonlinear function with respect to a high-dimensional noise (Equation 3). Thus we require likelihood-free methods, which assume that one can only sample from the model's likelihood (Marin et al., 2012; Tran et al., 2017). Here we apply likelihood-free variational inference (LFVI), which we scale to billions of genetic measurements (Tran et al., 2017).

As with all variational methods, LFVI posits a family of distributions over the latent variables and then optimizes to find the member closest to the posterior. For the variational family, we specify normal distributions with diagonal covariance for the SNP components $w_m$ and confounder $z_n$,

$$q(w_m) = \text{Normal}(w_m; \mu_{w_m}, \sigma_{w_m} I), \qquad q(z_n) = \text{Normal}(z_n; \mu_{z_n}, \sigma_{z_n} I).$$

We specify a point mass for the variational family on both neural network parameters $\phi$ and $\theta$. (This is equivalent to point estimation in a variational EM setting.)

For LFVI to scale to massive GWAS data, we use stochastic optimization by subsampling SNPs (Gopalan et al., 2016). At a high level, the algorithm proceeds in two stages. In the first stage, LFVI cycles through the following steps:

1. Sample SNP location $m$ and collect the observations at that location from all individuals.
2. Use the observations and current estimate of the confounders $z_{1:N}$ to update the $m^{th}$ SNP component $w_m$ and SNP neural network parameters $\phi$.
3. Use the observations and current estimate of SNP components $w_{1:M}$ to update the confounders $z_{1:N}$.

This first stage infers the posterior distribution of confounders $z_n$ and SNPs parameters $w_m$ and $\phi$. Each step's computation is independent of the number of SNPs, allowing us to scale to millions of genetic factors. In experiments, the algorithm converges while scanning over the full set of SNPs only once or twice.

In the second stage, we infer the posterior of outcome parameters $\theta$ given the inferred confounders from the first stage. Appendix C describes the algorithm in more detail; it expands on the LFVI implementation in Edward (Tran et al., 2016).

## 5 EMPIRICAL STUDY

We described implicit causal models, how to adjust for latent population-based confounders, and how to perform scalable variational inference. In general, validating causal inferences on observational data is not possible (Pearl, 2000). Therefore to validate our work, we perform an extensive simulation study on 100,000 SNPs, 940 to 5,000 individuals, and across 100 replications of 11 settings. The study indicates that our model is significantly more robust to spurious associations, with a state-of-the-art gain of 15-45.3% in accuracy. We also apply our model to a real-world GWAS of Northern Finland Birth Cohorts; our model indeed captures real causal relationships—identifying similar SNPs as previous state of the art.

We compare against three methods that are currently state of the art: PCA with linear regression (Price et al., 2006) ("PCA"); a linear mixed model (with the EMMAX software) (Kang et al., 2010) ("LMM"); and logistic factor analysis with inverse regression (Song et al., 2015) ("GCAT"). In all experiments, we use Adam with a initial step-size of 0.005, initialize neural network parameters uniformly with He variance scaling (He et al., 2015), and specify the neural networks for traits and SNPs as fully connected with two hidden layers, ReLU activation, and batch normalization (hidden layer sizes described below). For the trait model's neural network, we found that including latent variables as input to the final output layer improves information flow in the network.

### 5.1 SIMULATION STUDY: ROBUSTNESS TO SPURIOUS ASSOCIATIONS

We analyze 11 simulation configurations, where each configuration uses 100,000 SNPs and 940 to 5,000 individuals. We simulate 100 GWAS data sets per configuration for a grand total of 4,400 fitted models (4 methods of comparison). Each configuration employs a true model to generate the SNPs and traits based on real genomic data. Following Hao et al. (2016), we use the Balding-Nichols model based on the HapMap dataset (Balding & Nichols, 1995; Gibbs et al., 2003); PCA based on

| Trait | ICM | PCA (Price+06) | LMM (Kang+10) | GCAT (Song+10) |
|---|---|---|---|---|
| HapMap | **99.2** | 34.8 | 30.7 | **99.2** |
| TGP | **85.6** | 2.7 | 43.3 | 70.3 |
| HGDP | **91.8** | 6.8 | 40.2 | 72.3 |
| PSD ($a = 1$) | **97.0** | 80.4 | 92.3 | 95.3 |
| PSD ($a = 0.5$) | **94.3** | 79.5 | 90.1 | 93.6 |
| PSD ($a = 0.1$) | **92.2** | 38.1 | 38.6 | 90.4 |
| PSD ($a = 0.01$) | **92.7** | 24.2 | 35.1 | 90.7 |
| Spatial ($a = 1$) | **90.9** | 56.4 | 60.0 | 75.2 |
| Spatial ($a = 0.5$) | **86.2** | 50.5 | 46.6 | 72.5 |
| Spatial ($a = 0.1$) | **80.9** | 2.4 | 26.6 | 35.6 |
| Spatial ($a = 0.01$) | **75.5** | 1.8 | 15.3 | 30.2 |

**Table 1:** Precision accuracy over an extensive set of configurations and methods; we average over 100 simulations for a grand total of 4,400 fitted models. The setting $a$ in PSD and Spatial determines the amount of sparsity in the latent population structure: lower $a$ means higher sparsity. ICM is significantly more robust to spurious associations, outperforming other methods by up to 45.3%.

the 1000 Genomes Project (TGP) (Consortium et al., 2010); PCA based on the Human Genome Diversity project (HGDP) (Rosenberg et al., 2002); four variations of the Pritchard-Stephens-Donelly model (PSD) based on HGDP (Pritchard et al., 2000); and four variations of a configuration where population structure is determined by a latent spatial position of individuals. Only 10 of the 100,000 SNPs are set to be causal. Appendix D provides more detail.

Table 1 displays the precision for predicting causal factors across methods. When failing to account for population structure, "spurious associations" occur between genetic markers and the trait of interest, despite the fact that there is no biological connection. Precision is the fraction of the number of true positives over the number of true and false positives. This measures a method's robustness to spurious associations: higher precision means fewer false positives and thus more robustness.[1]

Table 1 shows that our method achieves state of the art across all configurations. Our method especially dominates in difficult tasks with sparse (small $a$), spatial (Spatial), and/or mixed membership structure (PSD): there is over a 15% margin in difference to the second best in general, and up to a 45.3% margin on the Spatial ($a = 0.01$) configuration. For simpler configurations, such as a mixture model (HapMap), our method has comparable performance.

## 5.2 NORTHERN FINLAND BIRTH COHORT DATA

We analyze a real-world GWAS of Northern Finland Birth Cohorts (Sabatti et al., 2009), which measure several metabolic traits and height and which contain 324,160 SNPs and 5,027 individuals. We separately fitted 10 implicit causal models, each of which models the effect of SNPs on one of ten traits. To specify the implicit causal models, we set the latent dimension of confounders to be 6 (following Song et al. (2015)). We use 512 units in both hidden layers of the SNP neural network and use 32 and 256 units for the trait neural network's first and second hidden layers respectively. Appendix E provides more detail.

Table 2 compares the number of identified causal SNPs across methods, with an additional "uncorrected" baseline, which does not adjust for any latent population structure. Each method is performed with a subsequent correction as measured by the genomic control inflation factor (Sabatti et al., 2009). Our models identify similar causal SNPs as previous methods. Interestingly, our model tends to agree with Song et al. (2015), identifying a total of 15 significant loci (Song et al. (2015) identified 16; others identified 11-14 loci). This makes sense intuitively, as Song et al. (2015) uses logistic factor analysis which, compared to all methods, most resembles our model.

---

[1]We also measured recall. All true-positives were found across all methods (like a real-world experiment, only few (10) SNPs are causal). Rarely did a method deviate and if so, it missed at most one.

| Trait | **ICM** | GCAT | LMM | PCA | Uncorrected |
|---|---|---|---|---|---|
| Body mass index | 0 | 0 | 0 | 0 | 0 |
| C-reactive protein | 2 | 2 | 2 | 2 | 2 |
| Diastolic blood pressure | 0 | 0 | 0 | 0 | 0 |
| Glucose levels | 3 | 3 | 2 | 2 | 2 |
| HDL cholesterol levels | 4 | 4 | 4 | 2 | 4 |
| Height | 1 | 1 | 0 | 0 | 0 |
| Insulin levels | 0 | 0 | 0 | 0 | 0 |
| LDL cholesterol levels | 3 | 4 | 3 | 3 | 3 |
| Systolic blood pressure | 0 | 0 | 0 | 0 | 0 |
| Triglyceride levels | 2 | 2 | 3 | 2 | 2 |

**Table 2:** Number of significant loci at genome-wide significance ($p < 7.2 \times 10^{-8}$) for each of the ten traits from NFBC data. The counts for GCAT are obtained from Song et al. (2015, Table 1); counts for LMM, PCA, and uncorrected are obtained from Kang et al. (2010, Table 2). The implicit causal model (**ICM**) captures causal relationships comparable to previously work.

## 6    DISCUSSION

We described implicit causal models, a rich class of models that can capture high-dimensional, nonlinear causal relationships. With genome-wide association studies, implicit causal models generalize previous successful methods to capture important nonlinearities, such as gene-gene and gene-population interaction. In addition, we described an implicit causal model that adjusts for confounders by sharing strength across examples. Our model achieves state-of-the-art accuracy, significantly outperforming existing genetics methods by 15-45.3%.

There are several limitations to learning true causal associations. For example, alleles at different loci typically exhibit linkage disequilibrium, which is a local non-random association influenced by factors such as the rate of recombination, mutation, and genetic drift. The implicit causal model might be extended with variables shared across subsets of SNPs to model the recombination process. Another limitation involves the data, where granularity of sequenced loci may lose signal or attribute causation to a region involving multiple SNPs. Better technology, and accounting for mishaps in the sequencing process in the model, can help.

While we focused on GWAS applications in this paper, we also believe implicit causal models have significant potential in other sciences: for example, to design new dynamical theories in high energy physics; and to accurately model structural equations of discrete choices in economics. We're excited about applications to these new domains, leveraging modern probabilistic modeling and causality to drive new scientific understanding.

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

## A  CONSISTENCY

Consider the simplest setting in § 2, where the causal graph is as shown with a global confounder with finite dimension and where we observe the data set $\{(x_n, y_n)\}$. Assume the specified model family over the causal graph includes the true data generating process.

First consider an atomic intevention $\text{do}(X = x)$ and let $\beta^*$ be the true structural value that generated our observations. The probability of a new outcome given the intervention and global structure is

$$p^*(y \mid \text{do}(X = x), \beta^*)) = p^*(y \mid x, \beta^*).$$

This follows from the backdoor criterion on the empty set. By Bernstein von-Mises, the posterior for our model $p(\beta \mid \mathbf{x}, \mathbf{y})$ concentrates at $\beta^*$. Thus, similarly, our posterior for $\theta$ given $\beta^*$ concentrates to the true functional mechanism $f_y$. This implies we have a consistent estimate of $p^*(y \mid \text{do}(\cdot), \beta^*)$.

This simple proof rigorizes the ideas behind learning and fixing population structure, a common heuristic in GWAS methods. Moreover, it lets us understand how to extend them to more complex latent variable models of the confounder and also provide uncertainty estimates of the latent structure become important as we apply these methods to finite data in practice.

## B  IMPLICIT CAUSAL MODEL IN EDWARD

We provide an example of an implicit causal model written in the Edward language below. It writes neural net weights and biases as model parameters. It does not include priors on the weights or biases; we add these as penalties to the objective function during training.

```
1   import edward as ed
2   import tensorflow as tf
3   from edward.models import Binomial, Normal
4
5   N = 5000   # number of individuals
6   M = 100000   # number of SNPs
7   K = 25   # latent dimension
8
9   def snp_neural_network(z, w):
10    z_tile = tf.tile(tf.reshape(z, [N, 1, K]), [1, M, 1])
11    w_tile = tf.tile(tf.reshape(w, [1, M, K]), [N, 1, 1])
12    h = tf.concat([z_tile, w_tile], 2)
13    h = tf.layers.dense(h, 512, activation=tf.nn.relu)
14    h = tf.layers.dense(h, 512, activation=tf.nn.relu)
15    h = tf.layers.dense(h, 1, activation=None)
16    return tf.reshape(h, [N, M])
17
18  def trait_neural_network(z, x):
19    eps = Normal(loc=0.0, scale=1.0, sample_shape=[N, 1])
20    h = tf.concat([z, x, eps], 1)
21    h = tf.layers.dense(h, 32, activation=tf.nn.relu)
22    h = tf.layers.dense(h, 256, activation=tf.nn.relu)
23    h = tf.concat([z, h], 1)   # include connection to z for output layer
24    h = tf.layers.dense(h, 1, activation=None)
25    return tf.reshape(h, [N])
26
27  z = Normal(loc=0.0, scale=1.0, sample_shape=[N, K])
28  w = Normal(loc=0.0, scale=1.0, sample_shape=[M, K])
29  logits = snp_neural_network(z, w)
30  x = Binomial(total_count=2.0, logits=logits)
31
32  # Note: Sometimes it's preferable to have neural net parameterize a
33  # distribution for tractable density such as for binary-valued traits.
34  # To do this, specify, e.g., `y = Bernoulli(logits=trait_neural_network(z, x))`
35  y = trait_neural_network(z, x)
```

## C   LIKELIHOOD-FREE VARIATIONAL INFERENCE FOR GWAS

### C.1   VARIATIONAL OBJECTIVE

The log-likelihood per-individual and per-SNP is

$$\log p(x_{nm}, y_n \,|\, w_m, z_n, \theta, \phi) = \log p(y_n \,|\, x_{n,1:M}, z_n, \theta) + \log p(x_{nm} \,|\, w_m, z_n, \phi).$$

There are local priors $p(w_m)$, $p(z_n)$ and global priors $p(\theta)$, $p(\phi)$. The posterior factorizes as

$$p(z_{1:N}, w_{1:M}, \phi, \theta \,|\, \mathbf{x}, \mathbf{y})$$
$$= p(\phi \,|\, \mathbf{x}) p(\theta \,|\, \mathbf{x}, \mathbf{y}) \prod_{m=1}^{M} p(w_m \,|\, x_{1:N,M}, \phi) \prod_{n=1}^{N} p(z_n \,|\, x_{n,1:M}, y_n, \theta, \phi, w_{1:M}).$$

Let the variational family follow the posterior's factorization above. For notational convenience, we drop the data dependence in $q$, $q(\phi) = q(\phi \,|\, \mathbf{x})$, $q(\theta) = q(\theta \,|\, \mathbf{x}, \mathbf{y})$ $q(w_m \,|\, \phi) = q(w_m \,|\, x_{1:N,m}, \phi)$, $q(z_n \,|\, \theta, \phi, w_{1:M}) = q(z_n \,|\, x_{n,1:M}, y_n, \theta, \phi, w_{1:M})$. We write the evidence lower bound and decompose the model's log joint density and the variational density,

$$\mathcal{L} = \sum_{m=1}^{M} \sum_{n=1}^{N} \mathbb{E}_{q(\phi)q(w_m \,|\, \phi)q(\theta)q(z_n \,|\, \theta, \phi, w_{1:M})} \Big[ \log p(x_{nm} \,|\, w_m, z_n, \phi) \Big] +$$
$$\sum_{n=1}^{N} \mathbb{E}_{q(\phi)q(w_{1:M} \,|\, \phi)q(\theta)q(z_n \,|\, \theta, \phi, w_{1:M})} \Big[ \log p(y_n \,|\, x_{n,1:M}, z_n, \theta) \Big] +$$
$$\sum_{m=1}^{M} \mathbb{E}_{q(\phi)q(w_m \,|\, \phi)} \Big[ \log p(w_m) - \log q(w_m \,|\, \phi) \Big] +$$
$$\sum_{n=1}^{N} \mathbb{E}_{q(\phi)q(w_{1:M} \,|\, \phi)q(\theta)q(z_n \,|\, \theta, \phi, w_{1:M})} \Big[ \log p(z_n) - \log q(z_n \,|\, \theta, \phi, w_{1:M}) \Big] +$$
$$\mathbb{E}_{q(\phi)} [\log p(\phi) - \log q(\phi)] + \mathbb{E}_{q(\theta)} [\log p(\theta) - \log q(\theta)].$$

Assume that the variational family for $z_n$ and $w_m$ are independent of other variables, $q(z_n)$ and $q(w_m)$. Also assume delta point masses for $q(\theta) = \mathbb{I}[\theta - \theta']$ parameterized by $\theta'$ and $q(\phi) = \mathbb{I}[\phi - \phi']$ parameterized by $\phi'$. This simplifies the objective, reducing to

$$\mathcal{L} = \sum_{m=1}^{M} \sum_{n=1}^{N} \mathbb{E}_{q(w_m)q(z_n)} \Big[ \log p(x_{nm} \,|\, w_m, z_n, \phi') \Big] + \sum_{n=1}^{N} \mathbb{E}_{q(z_n)} \Big[ \log p(y_n \,|\, x_{n,1:M}, z_n, \theta') \Big] +$$
$$\sum_{m=1}^{M} \mathbb{E}_{q(w_m)} \Big[ \log p(w_m) - \log q(w_m) \Big] + \sum_{n=1}^{N} \mathbb{E}_{q(z_n)} \Big[ \log p(z_n) - \log q(z_n) \Big] +$$
$$\log p(\phi') + \log p(\theta').$$

Each expectation can be unbiasedly estimated with Monte Carlo. For gradient-based optimization, we use reparameterization gradients (Rezende et al., 2014). We describe them next.

### C.2   FIRST STAGE: LEARNING THE CONFOUNDER

We provide details for the gradients. Let $\lambda_{w_m}$ parameterize $q(w_m; \lambda_{w_m})$ and $\lambda_{z_n}$ parameterize $q(z_n; \lambda_{z_n})$. We are interested in training the parameters $\lambda_{w_m}, \lambda_{z_n}, \theta', \phi'$. Subsample a SNP location $m \in \{1, \ldots, M\}$. Draw a sample $z_n' \sim q(z_n; \lambda_{z_n})$ for $n = 1, \ldots, N$ and $w_m' \sim q(w_m; \lambda_{w_m})$ for $m = 1, \ldots, M$, where the samples are reparameterizable (see Rezende et al. (2014) for details).

The gradient with respect to parameters $\lambda_{z_n}$ is unbiasedly estimated by

$$\nabla_{\lambda_{z_n}} \mathcal{L} \approx \nabla_{\lambda_{z_n}} \Big[ \log p(x_{nm} \,|\, w_m', z_n', \phi') + \log p(y_n \,|\, x_{n,1:M}, z_n', \theta') + \log p(z_n') - \log q(z_n') \Big].$$

This gradient scales linearly with the number of SNPs $M$. This is undesirable as the number of SNPs ranges from the hundreds of thousands to millions. We prevent this scaling by observing that for large $M$, the information in $x_{n,1:M}$ will influence the posterior far more than the single scalar $y_n$. In math, $p(z_n \mid x_{n,1:M}, y_n, \theta, \phi, w_{1:M}) \approx p(z_n \mid x_{n,1:M}, \theta, \phi, w_{1:M})$. This is a tacit assumption in all GWAS methods that adjust for the confounder (Yu et al., 2005; Price et al., 2006; Astle & Balding, 2009; Kang et al., 2010).

The gradient with respect to parameters $\lambda_{z_n}$ simplifies to

$$\nabla_{\lambda_{z_n}} \mathcal{L} \approx \nabla_{\lambda_{z_n}} \Big[ \log p(x_{nm} \mid w'_m, z'_n, \phi') + \log p(z'_n) - \log q(z'_n) \Big],$$

which scales to massive numbers of SNPs.

The gradients with respect to parameters $\lambda_{w_m}$ and $\phi'$ are unbiasedly estimated by

$$\nabla_{\lambda_{w_m}} \mathcal{L} \approx \sum_{n=1}^{N} \Big[ \nabla_{\lambda_{w_m}} \log p(x_{nm} \mid w'_m, z'_n, \phi') \Big] + \nabla_{\lambda_{w_m}} \log p(w'_m) - \nabla_{\lambda_{w_m}} \log q(w'_m),$$

$$\nabla_{\phi'} \mathcal{L} \approx \sum_{n=1}^{N} \nabla_{\phi'} \log p(x_{nm} \mid w'_m, z'_n, \phi').$$

Note how none of this depends on the trait $y$ or trait parameters $\theta$. We can thus perform inference to first approximate the posterior $p(z_{1:N}, w_{1:M}, \phi \mid \mathbf{x}, \mathbf{y})$. In a second stage, we can then perform inference to approximate the posterior $p(\theta \mid z_{1:N}, w_{1:M}, \mathbf{x}, \mathbf{y})$. The computational savings is significant not only within task but across tasks: when modelling many traits of interest (for example, § 5.2), inference over the SNP confounders only needs to be done once and can be re-used. We perform stochastic gradient ascent using these gradients to maximize the variational objective.

## C.3   SECOND STAGE: LEARNING THE TRAIT

Above we described the first stage of an algorithm which performs stochastic gradient ascent to optimize parameters so that $q(z_n)q(w_{1:M})q(\theta') \approx p(z_{1:N}, w_{1:M}, \phi \mid \mathbf{x}, \mathbf{y})$. Given these parameters, we are interested in training $\theta'$. Dropping constants with respect to $\theta$ in the objective, we have

$$\mathcal{L} \propto \sum_{n=1}^{N} \mathbb{E}_{q(z_n)} \Big[ \log p(y_n \mid x_{n,1:M}, z_n, \theta') \Big] + \log p(\theta').$$

We maximize this objective using stochastic gradients with a single sample $z'_n \sim q(z_n)$,

$$\nabla_{\theta'} \mathcal{L} \approx \sum_{n=1}^{N} \nabla_{\theta'} \log p(y_n \mid x_{n,1:M}, z'_n, \theta') + \nabla_{\theta'} \log p(\theta')$$

This corresponds to Monte Carlo EM. Its primary computation per-iteration is the backward pass of the trait's neural network. Unlike in the first stage, we do not subsample SNPs as the likelihood depends on all SNPs.

## C.4   SECOND STAGE: HANDLING LIKELIHOOD-FREE TRAITS

In general, the density of $y_n$ is intractable: we exploit its tractable density if $y_n$ is discrete (it induces a categorical likelihood); otherwise for real-valued traits, we perform likelihood-free inference with respect to $y_n$. Following LFVI (Tran et al., 2017), define $q(y)$ to be the empirical distribution over observed data $\{y_n\}$. Then subtract it as a constant to the objective, so

$$\mathcal{L} \propto \sum_{n=1}^{N} \mathbb{E}_{q(z_n)} \Big[ \log p(y_n \mid x_{n,1:M}, z_n, \theta') - \log q(y_n) \Big] + \log p(\theta').$$

We approximate this log-ratio with a ratio estimator, $r(y_n, x_{n,1:M}, z_n, \theta'; \lambda_r)$. It is a function of all inputs in the log-ratio and is parameterized by $\lambda_r$.

We train the ratio estimator by minimizing a loss function with respect to its parameters,

$$\mathbb{E}_{p(y_n \mid x_{n,1:M}, z_n, \theta')}[- \log \sigma(r(y_n, x_{n,1:M}, z_n, \theta'; \lambda_r))] + \mathbb{E}_{q(y_n)}[- \log(1 - \sigma(r(y_n, x_{n,1:M}, z_n, \theta'; \lambda_r)))].$$

The global minima of this objective with respect to the ratio estimator is the desired log-ratio,

$$r^*(y_n, x_{n,1:M}, z_n, \theta') = \log p(y_n \,|\, x_{n,1:M}, z_n, \theta') - \log q(y_n).$$

Unfortunately, the ratio estimator has inputs of many dimensions. In particular, it has the problematic property of scaling with the number of SNPs, which can be on the order of hundreds of thousands.

We can efficiently parameterize the ratio estimator by studying two extreme cases with respect to computational efficiency and statistical efficiency. In one extreme, suppose $y_n$ has a tractable Gaussian density with mean given by the outcome model's neural network and unit variance (that is, the neural net is parameterized to apply a location-shift on the noise input, $y_n = \text{NN}(\cdot) + \epsilon_n$). Up to additive and multiplicative constants, the optimal log-ratio is

$$r^*(y_n, x_{n,1:M}, z_n, \theta') \propto (y_n - \text{NN}([x_{n,1:M}, z_n] \,|\, \theta'))^2.$$

This implies the ratio estimator must relearn the neural network's forward pass in order to estimate the optimal log-ratio. This is computationally redundant and can lead to unstable training. On another extreme, suppose we parameterize $r$ as

$$r(y_n, x_{n,1:M}, z_n, \theta'; \lambda_r) = r(y_n, \text{NN}([x_{n,1:M}, z_n, \epsilon] \,|\, \theta'); \lambda_r).$$

This dramatically reduces $r$'s input dimensions, from hundreds of thousands to just two. However, while computationally efficient, this is a poor statistical approximation: there is only a single dimension to preserve information about $x_{n,1:M}$, $z_n$, $\epsilon_n$, and $\theta'$ relevant to $y_n$; this dimension is lossy for even Gaussian densities.

As a middleground, we use the neural net's first hidden layer as input into the ratio estimator,

$$r(y_n, x_{n,1:M}, z_n, \theta'; \lambda_r) = r(y_n, h_n; \lambda_r).$$

This reduces the ratio estimator's inputs to be the trait $y_n$ and first hidden layer's units $h_n$. This hidden layer has much fewer dimensions than the raw inputs, such as 32 units (making it computationally efficient). Moreover, under the data processing inequality (Cover & Thomas, 1991), $h_n$ preserves more information relevant to $y_n$ than subsequent layers of the neural network (making it statistically efficient). For all experiments, we parameterized $r$ with two fully connected hidden layers with equal number of hidden units.

The gradient with respect to parameters $\theta$ is estimated by

$$\nabla_{\theta'} \mathcal{L} \approx \sum_{n=1}^{N} \nabla_{\theta'} r(y_n, x_{n,1:M}, z_n', \theta'; \lambda_r) + \log p(\theta').$$

This substitutes in the ratio estimator as a proxy to the intractable likelihood.

We minimize the auxiliary loss function in order to train the ratio estimator. Sample $y_n' \sim p(y_n \,|\, x_{n,1:M}, z_n, \theta')$ and subsample a data point $y_n \sim q(y_n)$. The gradient is estimated by

$$\nabla_{\lambda_r} \cdot \approx \nabla_{\lambda_r} \Big[ -\log \sigma(r(y_n', h_n; \lambda_r)) - \log(1 - \sigma(r(y_n, h_n; \lambda_r))) \Big].$$

This corresponds to maximum likelihood, balanced with an adversarial objective to estimate the likelihood, and is relatively fast. We perform stochastic gradient ascent, alternating between these two sets of gradients.

## D    SIMULATION STUDY

We provide more detail to § 5.1. Implicit causal models can not only represent many causal structures but, more importantly, learn them from data. To demonstrate this, we simulate data from a comprehensive collection of popular models in GWAS and analyze how well the fitted model can capture them. These configurations exactly follow Hao et al. (2016) with same hyperparameters, which we describe below.

For each of the 11 simulation configurations, we generate 100 independent data sets. Each data set consists of a $M \times N$ matrix of genotypes $X$ and vector of $N$ traits $y$. Each individual $n$ has $M$ SNPs and one trait.

## D.1    GENOTYPE MATRIX

To simulate the $M \times N$ matrix of genotypes $X$, we draw $x_{mn} \sim \mathrm{Binomial}(2, \pi_{mn})$ for $m = 1, \ldots, M$ SNPs and $n = 1, \ldots, N$ individuals. The probabilities $\pi_{mn}$ can be encoded under a real-valued $M \times N$ matrix of allele frequencies $F$ where $\pi_{mn} = [\mathrm{logit}(F)]_{mn}$.

Many models in GWAS can be described under the factorization $F = \Gamma S$, where $\Gamma$ is a $M \times K$ matrix and $S$ is a $K \times N$ matrix for a fixed rank $K \leq N$. This includes principal components analysis (Price et al., 2006), the Balding-Nichols model (Balding & Nichols, 1995), and the Pritchard-Stephens-Donnelly model (Pritchard et al., 2000). The $M \times K$ matrix $\Gamma$ describes how structure manifests in the allele frequencies across SNPs. The $K \times N$ matrix $S$ encapsulates the genetic population structure across individuals.

We describe how we form $\Gamma$ and $S$ for each of the 11 simulation configurations.

**Balding-Nichols Model (BN) + HapMap.**    The BN model describes individuals according to a discrete mixture of ancestral subpopulations (Balding & Nichols, 1995). The HapMap data set was collected from three discrete populations (Gibbs et al., 2003), which allows us to populate each row $m$ of $\Gamma$ with three i.i.d. draws from the Balding-Nichols model: $\Gamma_{mk} \sim \mathrm{BN}(p_m, F_m)$, where $k \in \{1, 2, 3\}$. Each $\Gamma_{mk}$ is interpreted to be the allele frequency for subpopulation $k$ at SNP $m$. The pairs $(p_m, F_m)$ are computed by randomly selecting a SNP in the HapMap data set, calculating its observed allele frequency, and estimating its $F_{ST}$ value using the estimator of Weir & Cockerham (1984). The columns of $S$ are populated with indicator vectors such that each individual is assigned to one of the three subpopulations. The subpopulation assignments are drawn independently with probabilities $60/210$, $60/210$, and $90/210$, which reflect the subpopulation proportions in the HapMap data set. The simulated data has $M = 100,000$ SNPs and $N = 5000$ individuals.

**1000 Genomes Project (TGP).**    TGP is a project that comprehensively catalogs human genetic variation by producing complete genome sequences of well over 1000 individuals of diverse ancestries (Consortium et al., 2010). To form $\Gamma$, we sample $\Gamma_{mk} \sim 0.9\,\mathrm{Uniform}(0, 1/2)$ for $k = 1, 2$ and set $\Gamma_{m3} = 0.05$. To form $S$, we compute the first two principal components of the TGP genotype matrix after mean centering each SNP. We then transform each principal component to be between $(0, 1)$ and set the first two rows of $S$ to be the transformed principal components. The third row of $S$ is set to 1 as an intercept. The simulated data has $M = 100,000$ SNPs and $N = 1500$ individuals (the total number of individuals in the TGP data set).

**Human Genome Diversity Project (HGDP).**    HGDP is an international project that has genotyped a large collection of DNA samples from individuals distributed around the world, aiming to assess worldwide genetic diversity at the genomic level (Rosenberg et al., 2002). We followed the same scheme as for TGP above. The simulated data has $M = 100,000$ SNPs and $N = 940$ individuals (the total number of individuals in the HGDP data set).

**Pritchard-Stephens-Donnelly (PSD) + HGDP.**    The PSD model describes individuals according to an admixture of ancestral subpopulations (Pritchard et al., 2000). The rows of $\Gamma$ are drawn from three i.i.d. draws from the Balding-Nichols model: $\Gamma_{mk} \sim \mathrm{BN}(p_m, F_m)$, where $k \in \{1, 2, 3\}$. The pairs $(p_m, F_m)$ are computed by randomly selecting a SNP in the HGDP data set, calculating its observed allele frequency, and estimating its $F_{ST}$ value using the estimator of Weir & Cockerham (1984). The estimator requires each individual to be assigned to a subpopulation, which are made according to the $K = 5$ subpopulations from the analysis in Rosenberg et al. (2002). The columns of $S$ are sampled $(s_{1n}, s_{2n}, s_{3n}) \sim \mathrm{Dirichlet}(\boldsymbol{\alpha} = (a, a, a))$ for $n = 1, \ldots, N$. We apply four PSD configurations with hyperparameter settings of $a = 0.01, 0.1, 0.5, 1$. Varying $a$ from 1 to 0 varies the level of sparsity as individuals are placed from uniformly to corners of the simplex. The simulated data has $M = 100,000$ SNPs and $N = 5000$ individuals.

**Spatial.**    In this setting, we simulate genotypes such that the population structure relates to the spatial position of individuals. The matrix $\Gamma$ is populated by sampling $\Gamma_{mk} \sim 0.9\,\mathrm{Uniform}(0, 1/2)$ for $k = 1, 2$ and setting $\Gamma_{m3} = 0.05$. The first two rows of $S$ correspond to coordinates for each individual on the unit square and are set to be independent and identically distributed samples from $\mathrm{Beta}(a, a)$, while the third row of $S$ is set to 1 as an intercept. We apply four spatial configurations with hyperparameter settings of $a = 0.01, 0.1, 0.5, 1$. As with the Dirichlet distribution in the PSD model, varying $a$ from 1 to 0 varies the level of sparsity as individuals are placed from

uniformly to corners of the unit square. The simulated data has $M = 100,000$ SNPs and $N = 5000$ individuals.

## D.2   TRAITS OF INTEREST

To simulate traits $y$, we simulate from a linear model: for each individual $n$'s SNPs $\{x_{mn}\}$,

$$y_n = \sum_{m=1}^{M} \beta_m x_{mn} + \lambda_n + \epsilon_n, \qquad \epsilon_n \sim \text{Normal}(0, \sigma_n^2).$$

Each trait is real-valued and is determined by a linear combination of SNPs, a per-individual offset, and heteroskedastic noise. Below we describe how we set $\{\beta_m\}$, $\{\lambda_n\}$, and $\{\sigma_n\}$.

Without loss of generality, we set the first 10 SNPs to be true alternative SNPs ($\beta_m \neq 0$), where $\beta_m \sim \text{Normal}(0, 0.5)$ for $m = 1, 2, \ldots, 10$; $\beta_m = 0$ for $m > 10$. In order to simulate $\lambda_j$ and $\epsilon_j$ so that they are also influenced by the latent variables $z_1, \ldots, z_n$, we performed the following:

1. Run $K$-means clustering on the columns of $S$ with $K = 3$ using Euclidean distance. This assigns each individual $j$ to one of three partitions $S_1, S_2, S_3$ where $S_k \subset 1, 2, \ldots, n$.
2. Set $\lambda_j = k$ for all $j \in S_k$ for each $k = 1, 2, 3$.
3. Draw $\tau_1^2, \tau_2^2, \tau_3^2 \sim \text{InverseGamma}(3, 1)$. Set $\sigma_j = \tau_k$ for all $j \in S_k$.

Following Song et al. (2015),

> This strategy simulates non-genetic effects and random variation that manifest among $K$ discrete groups over a more continuous population genetic structure defined by $S$. This is meant to emulate the fact that environment (specifically lifestyle) may partition among individuals in a manner distinct from, but highly related to population structure.

We apply this strategy for each of the 11 configurations where each involved up to $M = 100,000$ SNPs and $N = 5000$ individuals. For each configuration, we simulated 100 independent data sets, thus requiring a total of 1100 fitted models per method of comparison.

## D.3   OTHER DETAILS

For testing, we obtain one $p$-value for each SNP, which is the probability that the SNP's effect on the trait of interest is zero. To do so, we apply a likelihood ratio test statistic for each SNP $m$,

$$T(\mathbf{x}, \mathbf{y}) = 2\left( \max_{\theta} \log p(\mathbf{y} \mid \mathbf{x}; \theta) - \max_{\theta_{-m}} \log p(\mathbf{y} \mid \mathbf{x}; \theta_{-m}, \theta_m = 0) \right)$$

where $\theta_m$ corresponds to weights connecting $x_m$ to hidden units in the trait neural networks' first layer. When $\theta_m = 0$ in the model, following Wilks' theorem, the null distribution of this test statistic is $\chi_H^2$ where $H$ is the number of first layer hidden units, i.e., the test statistic converges to this distribution as the number of individuals $N \to \infty$. This statistic and null distribution is a simple extension of Song et al. (2015), which applies a linear model. (See also their proof for the null distribution.)[2]

In summary, the testing procedure works as follows:

1. Formulate and estimate a model of population structure that provides well-behaved estimates of the posterior $p(z \mid \mathbf{x}, \mathbf{y})$.
2. First, perform a regression of the trait given SNPs and estimated posterior $p(z \mid \mathbf{x}, \mathbf{y})$. Second, for each SNP, perform a regression of the trait given SNPs and estimated posterior $p(z \mid \mathbf{x}, \mathbf{y})$, but with $\theta_m$ fixed at 0.
3. Calculate a $p$-value for each SNP. Use the test statistic above, which follows a $\chi_H^2$ null distribution for large sample sizes.

---

[2]The test only holds approximately for two reasons. First, it assumes asymptotically large sample size (as all frequentist tests). Second, given a multi-layered neural network, there exist permutations of the weights and biases beyond the first layer weights which set the effect of $x_m$ on $y$ to 0.

The $p$-value threshold is fixed to $t = 0.0025$ across all methods (Song et al., 2015). To calculate the number of observed positives, we count the number of $p$-values for that are less than or equal to $t$. The true positives are the subset of $p$-values associated with causal SNPs; false positives are those associated with null SNPs.

Spurious associations occur when $p$-values corresponding to null SNPs are artificially small. Namely, false positives are spurious associations. In general, we expect there to be $m_0 \times t$ false positives among the $m_0$ $p$-values corresponding to null SNPs; in our setting, this corresponds to $(100,000 - 10) \cdot 0.0025 \approx 250$ SNPs. A method properly accounts for structure when the average excess is no more than this number. Our precision count involved only the number of false positives higher than this calculation (which depends on the number of SNPs in that setting).

To specify the implicit causal model in our experiments, we set the latent dimension of confounders equal to 3 or 5. We use 512 units in both hidden layers of the SNP neural network and use 32 and 256 units for the trait neural network's first and second hidden layers respectively.

## E  Northern Finland Birth Cohort Data

We provide more detail to § 5.2. The data was obtained from the database of Genotypes and Phenotypes (dbGaP) (`phs000276.v1.p1`). We follow the same preprocessing as Song et al. (2015),

> Individuals were filtered for completeness (maximum 1% missing genotypes) and pregnancy. (Pregnant women were excluded because we did not receive IRB approval for these individuals.) SNPs were first filtered for completeness (maximum 5% missing genotypes) and minor allele frequency (minimum 1% minor allele frequency), then tested for Hardy-Weinberg equilibrium (p-value < 1/328348). The final dimensions of the genotype matrix are $m = 324,160$ SNPs and $n = 5027$ individuals.

> A Box-Cox transform was applied to each trait, where the parameter was chosen such that the values in the median 95% value of the trait was as close to the normal distribution as possible. Indicators for sex, oral contraception, and fasting status were added as adjustment variables. For glucose, the individual with the minimum value was removed from the analysis as an extreme outlier.

No additional changes were made to the data.

After fitting each model, we follow the same procedure as in the simulation study for predicting causal factors. We set the $p$-value threshold to be the genome-wide threshold of $7.2 \times 10^{-8}$ following Kang et al. (2010).

