# OpenReview forum: "Implicit Causal Models for Genome-wide Association Studies"
_ICLR.cc/2018/Conference — Accept (Poster)_

### Official Review · AnonReviewer3 · 2017-11-25
**Interesting idea but needs more experiments and justification since it is a vast field and not all aspects of the problem is accounted for.**

**Rating:** 5
**Confidence:** 5

**Review:**

In this paper, the authors propose to use the so-called implicit model to tackle Genome-Wide Association problem. The model can be viewed as a variant of Structural Equation Model. Overal the paper is interesting and relatively well-written but some important details are missing and way more experiments need to be done to show the effectiveness of the approach.

*  How do the authors call a variant to be associated with the phenotype (y)? More specifically, what is the distribution of the null hypothesis? Section D.3 in the appendix does not explain the hypothesis testing part well. This method models $x$ (genetic), $y$ (phenotype), and $z$ (confounder) but does not have a latent variable for the association. For example, there is no latent indicator variable (e.g., Spike-Slab models [1]) for each variant.  Did they do hypothesis testing separately after they fit the model? If so, this has double dipping problem because the data is used once to fit the model and again to perform statistical inference.

* In GWAS, a method resulting in high power with control of FP is favored. In traditional univariate GWAS, the false positive rate is controlled by genome-wide significant level (7e-8), Bonferroni correction or other FP control approaches.  Why Table 1 does not report FP? I need Table 1 to report the following: What is the power of this method if FPR is controlled(False Positive Rate < 0.05)? Also, the ROC curve for FPR<0.05 should be reported for all methods.

* I believe that authors did a good job in term of a survey of the available models for GWA from marginal regression to mixed effect model, etc. The authors account for typical confounders such as cryptic relatedness which I liked. However, I recommend the authors to be cautious calling the association detected by their method " a Causal Association." There are tons of research done to understand the causal effect of the genetic variants and this paper (and this venue) is not addressing those.  There are several ways for an associated variant to be non-causal and this paper does not even scratch the surface of that. For example, in many studies, discovering the causal SNPs means finding a genetic variant among the SNPs in LD of each other (so-called fine mapping). The LD-pruning procedure proposed in this paper does not help for that purpose.

* This approach jointly models the genetic variants and the phenotype (y). Let us assume that one can directly maximize the ML (ELBO maximizes a lower bound of ML). The objective function is disproportionally influenced by the genetic variants (x) than y because M is very large (  $\prod_{m=1}^M p(w) p(x|z,w,\phi)   >>   p(z) p(y|x,z,\theta) $  ). Effectively, the model focuses on the genetic variants, not by the disease. This is why multi-variate GWAS focuses on the conditional p(y|x,z) and not p(y,x,z). Nothing was shown in the paper that this focusing on p(y,x,z) is advantageous to p(y|x,z).

* In this paper, the authors use deep neural networks to model the general functional causal models. Since estimation of the causal effects is generally unidentifiable (Sprites 1993), I think using a general functional causal model with confounder modeling would have a larger chance to weaken the causal effects because the confounder part can also explain part of the causal influences. Is there a theoretical guarantee for the proposed method? Practically, how did the authors control the model complexity to avoid trivial solutions?

Minor
-------
* The idea of representing (conditional) densities by neural networks was proposed in the generative adversarial networks (GAN). In this paper, the authors represent the functional causal models by neural networks, which is very related to the representation used in GANs. The only difference is that GAN does not specify a causal interpretation. I suggest the authors add a short discussion of the relations to GAN.

* Previous methods on causal discovery rely on restricted functional causal models for identifiability results. They also use Gaussian process or multi-layer perceptron to model the functions implicitly, which can be consider as neural networks with one hidden layer. The sentence “These models typically focus on the task of causal discovery, and they assume fixed nonlinearities or smoothness which we relax using neural networks.” in the related work section is not appropriate.

[1] Scalable Variational Inference for Bayesian Variable Selection in Regression, and Its Accuracy in Genetic Association Studies

---

> ### Author Response · Authors · 2018-01-05
> **re:AnonReviewer3**
>
> Thanks for the details, suggestions, and praise! On the experimental validation, see discussion in the Comment to all reviewers.
>
> > * However, I recommend the authors to be cautious calling the association detected by their method " a Causal Association." [...]
>
> Thanks for this caution. We added these limitations and notes in the revision. In short, we agree guaranteeing real-world causation has myriads of complexity. For example, the discussion now includes future work to incorporate linkage disequilibrium and the granularity of SNP loci.
>
> > * [...] Effectively, the model focuses on the genetic variants, not by the disease. This is why multi-variate GWAS focuses on the conditional p(y|x,z) and not p(y,x,z). Nothing was shown in the paper that this focusing on p(y,x,z) is advantageous to p(y|x,z).
>
> This might be a misunderstanding: we also focus on the conditional, as the goal is to infer parameters from p(y | x). We only use the joint p(y, x, z) to adjust for population-confounders, and which is similar to common techniques in GWAS. The paragraph below Eq 4 describes that like two-stage estimation, the method can be thought of as first inferring p(z | x, y); then it infers parameters in p(y | x, z), drawn over posterior samples of z.
>
> > * [...] Since estimation of the causal effects is generally unidentifiable (Sprites 1993), I think using a general functional causal model with confounder modeling would have a larger chance to weaken the causal effects because the confounder part can also explain part of the causal influences. Is there a theoretical guarantee for the proposed method? Practically, how did the authors control the model complexity to avoid trivial solutions?
>
> Proposition 1 provides a consistency guarantee, rendering the adjustment for latent confounders valid in observational data (assuming the causal graph is correct).  In practice, the number of latent dimension in the confounder reduces to typical probabilistic modeling with latent variable models, precisely like the number of latent factors in PCA (Price et al., 2006) and logistic factor analysis (Song, Hao, Storey, 2015). In our experiments, Appendix D explains that we fix the latent dimension across methods and experiments.
>
> > * The idea of representing (conditional) densities by neural networks was proposed in the generative adversarial networks (GAN). In this paper, the authors represent the functional causal models by neural networks, which is very related to the representation used in GANs. [...]
>
> Thanks for the note. We avoided discussion of GANs as we did not use any of their architecture or training insights. However, implicit models are indeed a model class including GANs (e.g., Mohamed and Lakshminarayanan, 2016). We also applied likelihood-free variational inference (Tran et al., 2017), which uses adversarial training.
>
> > * Previous methods on causal discovery rely on restricted functional causal models for identifiability results. They also use Gaussian process or multi-layer perceptron to model the functions implicitly, which can be consider as neural networks with one hidden layer. [...]
>
> Do you have references for functional causal models with MLPs? We'd love to revise the statement for models not assuming fixed nonlinearities. We're only familiar with works such as Mooij et al. (2010), which uses GPs and does typically assume smoothness via its kernel. In private discussion with Joris Mooij, we have clarified the statement.

---

### Official Review · AnonReviewer2 · 2017-11-27
**good paper, unclear audience**

**Rating:** 6
**Confidence:** 5

**Review:**

This paper tackles two problems common in genome-wide association studies: confounding (i.e. structured noise) due to population structure and the potential presence of non-linear interactions between different parts of the genome. To solve the first problem this paper effectively suggests learning the latent confounders jointly with the rest of the model. For the second problem, this paper proposes “implicit causal models’, that is, models that  leverage neural architectures with an implicit density.

The main contribution of this paper is to create a bridge between the statistical genetics community and the ML community. The method is technically sound and does indeed generalize techniques currently used in statistical genetics. The main concerns with this paper is that 1) the claim that it can detect epistatic interactions is not really supported. Yes, in principle the neural model used to model y could detect them, but no experiments are shown to really tease this case apart 2) validating GWAS results is really hard, because no causal information is usually available. The authors did a great job on the simulation framework, but table 1 falls short in terms of evaluation metric: to properly assess the performance of the method on simulated data, it would be good to have evidence that the type 1 error is calibrated (e.g. by means of qq plots vs null distribution) for all methods. At the very least, a ROC curve could be used to show the quality of the ranking of the causal SNPs for each method, irrespective of p-value cutoff.

Quality: see above. The technical parts of this paper are definitely high-quality, the experimental side could be improved.
Clarity: if the target audience of this paper is the probabilistic ML community, it’s very clear. If the statistical genetics community is expected to read this, section 3.1 could result too difficult to parse. As an aside: ICLR might be the right venue for this paper given the high ML content, but perhaps a bioinformatics journal would be a better fit, depending on intended audience.

---

> ### Author Response · Authors · 2018-01-05
> **re:AnonReviewer2**
>
> Thanks for the comments!
>
> > 1) the claim that it can detect epistatic interactions is not really supported. Yes, in principle the neural model used to model y could detect them, but no experiments are shown to really tease this case apart
>
> One quantitative evidence is that for all setting configurations excluding Spatial, the GCAT baseline captures the latent confounder as well as the implicit causal model (true data is generated from a class that the GCAT subsumes). This means the baseline and implicit causal model only differ by the trait's model, p(y | x, z). The baseline uses a linear model; the implicit causal model uses a neural network. The latter outperforms across all configurations.
>
> We're open to other suggestions on how to show this quantitatively. In practice, we also see that each trained hidden unit in the first layer has nonzero weights coming from multiple SNPs simultaneously.
>
> > 2) to properly assess the performance of the method on simulated data, it would be good to have evidence that the type 1 error is calibrated [...]
>
> See discussion in the Comment to all reviewers.
>
> > Clarity: if the target audience of this paper is the probabilistic ML community, it’s very clear. If the statistical genetics community is expected to read this, section 3.1 could result too difficult to parse.
>
> Thanks for the note! We targeted the audience to probabilistic ML. We're planning to submit another work applying these methods to new GWAS. There, we show newly discovered causal SNPs for the first time, and we believe this is more appropriate and interesting for the genetics community.

---

> > ### Comment · AnonReviewer2 · 2018-01-23
> > **experiments still need some work**
> >
> > The revision clarifies some of my concerns but the fact that all the methods get perfect recall (at which pv threshold?) made me look more into the simulation settings. 10 causal variants with weights drawn from N(0, 0.5) is an incredible amount of signal, making the problem too easy. For future revision of the paper I encourage the authors to sample the number of causal variants in [5, 2000] and make sure that the *total* variance explained by all the causal variants is within a reasonable range (I would say 0.01 to 0.1, but it’s arguable). I also encourage the authors to add Manhattan plots of both the simulated and real data as supplementary, alongside a qq-plot of the test statistics on null-only data on simulation (\beta_m = 0 for all m). Again: this is potentially a good paper, but the experiments need some work.

---

> > > ### Comment · AnonReviewer3 · 2018-01-24
> > > **Absolutely on point!**
> > >
> > > I totally agree with this reviewer. The effect size of the SNPs (unless they very penetrant) is 1%-5%. I even think 10% is too high. If the p-values were inflated, you can see that in the Q-Q plot. I also don't understand why ROC curve and more particularly for FP<5% (which is a knee of the ROC curve). This is what I asked earlier.
> > >
> > > I agree with the reviewer, good work but more experiment in a realistic setting is required to actually evaluate this method.

---

> > > ### Author Response · Authors · 2018-01-24
> > > **Running proposed experiments now**
> > >
> > > Note we followed the precise protocol of Song, Hao, Storey (2015).  We think it's important to stress this as we did not at all deviate from published experiments. Namely, we did not make any changes to the hyperparameters to be fair with baselines. (In fact, the experiment setup favors logistic factor analysis, not ours).
> > >
> > > We're running your proposed experiment now: # causal variants ~ [5, 2000], constraining variance of causal variants. We will update you in the next day or two.

---

> > > > ### Comment · AnonReviewer2 · 2018-01-25
> > > > **something to keep in mind**
> > > >
> > > > Song, Hao and Storey (2015) rescale the variance components  (Suppl information section 6 https://media.nature.com/original/nature-assets/ng/journal/v47/n5/extref/ng.3244-S1.pdf) and it's not clear from the paper that you do as well. So before you set the experiments running, I would suggest checking that you have rescaling "on". This can make a HUGE difference (from 50% of signal variance down to 5%).

---

> > > > > ### Author Response · Authors · 2018-01-26
> > > > > **re:something to keep in mind**
> > > > >
> > > > > Unfortunately we weren't able to finish the experiments by today, which is the deadline. Regardless if the paper is accepted, we hope to finish these experiments and get them into the paper by camera-ready and/or the next arxiv update. (And thanks again to all the reviewers for the helpful feedback.)
> > > > >
> > > > > re:rescale. To clarify, we used their precise code for the simulations and GCAT fitting (https://github.com/StoreyLab/gcatest). So we did rescale the variance components.

---

### Official Review · AnonReviewer1 · 2017-11-29
**The paper is overall well-written and makes new and non-trivial contributions to model inference and the application. However, not all claims are well-supported by the data provided in the paper.**

**Rating:** 6
**Confidence:** 5

**Review:**

The paper presents a non-linear generative model for GWAS that models population structure.
Non-linearities are modeled using neural networks as non-linear function approximators and inference is performed using likelihood-free variational inference.
The paper is overall well-written and makes new and non-trivial contributions to model inference and the application.
Stated contributions are that the model captures causal relationships, models highly non-linear interactions between causes and accounts for confounders. However, not all claims are well-supported by the data provided in the paper.
Especially, the aspect of causality does not seem to be considered in the application beyond a simple dependence test between SNPs and phenotypes.

The paper also suffers from unconvincing experimental validation:
- The evaluation metric for simulations based on precision is not meaningful without reporting the recall at the same time.

- The details on how significance in each experiment has been determined are not sufficient.
From the description in D.3 the p-value a p-value threshold of 0.0025 has been applied. Has this threshold been used for all methods?
The description in D.3 seems to describe a posterior probability of the weight being zero, instead of a Frequentist p-value, which would be the probability of estimating a parameter at least as large on a data set that had been generated with a 0-weight.

- Genomic control is applied in the real world experiment but not on the simulations. Genomic control changes the acceptance threshold of each method in a different way. Both precision and recall depend on this acceptance threshold. Genomic control is a heuristic that adjusts for being too anti-conservative, but also for being too conservative, making it hard to judge the performance of each method on its own. Consequently, the paper should provide additional detail on the results and should contrast the performance of the method without the use of genomic control.

minor:

The authors claim to model nonlinear, learnable gene-gene and gene-population interactions.
While neural networks may approximate highly non-linear functions, it still  seems as if the confounders are modeled largely as linear. This is indicated by the fact that the authors report performance gains from adding the confounders as input to the final layer.

The two step approach to confounder correction is compared to PCA and LMMs, which are stated to first estimate confounders and then use them for testing.
For LMMs this is not really true, as LMMs treat the confounder as a latent variable throughout and only estimate the induced covariance.

---

> ### Author Response · Authors · 2018-01-05
> **re:AnonReviewer1**
>
> Thanks for the detailed comments!  On the experimental validation, see discussion in the Comment to all reviewers.
>
> > The authors claim to model nonlinear, learnable gene-gene and gene-population interactions. While neural networks may approximate highly non-linear functions, it still seems as if the confounders are modeled largely as linear.
>
> Nonlinear effect strictly from confounder to trait was not necessary in practice. However, as evidenced by the experiments, the trait's neural network gets noticeable improvement from nonlinear interaction between genes, and between gene-confounder (population). (See reply to AnonReviewer2 for more details.)
>
> > The two step approach to confounder correction is compared to PCA and LMMs, which are stated to first estimate confounders and then use them for testing. For LMMs this is not really true, as LMMs treat the confounder as a latent variable throughout and only estimate the induced covariance.
>
> Thanks for this correction; we adjusted the comment in the revision.

---

### Author Response · Authors · 2018-01-05
**To all AnonReviewers (thanks!)**

Thanks to the three reviewers for their excellent feedback. They all found the paper interesting, well-written, and novel. Quoting R2 for example, "The main contribution of this paper is to create a bridge between the statistical genetics community and the ML community. The method is technically sound and does indeed generalize techniques currently used in statistical genetics."

We addressed comments in the replies and revision. All reviewers asked questions about the experiments; we provide more detail here.

> R1:  From the description in D.3 the p-value threshold of 0.0025 has been applied. Has this threshold been used for all methods?

The experiment's procedure strictly follows Song, Hao, Storey (2015). Namely, the p-value threshold is 0.0025 and set across all methods.

> R1:  [...]  the paper should provide additional detail on the results and should contrast the performance of the method without the use of genomic control.

We used genomic control only to compare to baselines in the literature for the real-world experiment. Unfortunately, we're unable to reproduce these papers' results using the genomic control. This makes it difficult to compare to baselines without genomic control unless we're unfair against them. It's also difficult to necessarily assess which method performs best as there is no ground truth: we can only establish that our work can indeed capture well-recognized causal SNPs as the baselines.

> R1, R2, and R3 ask about recall, false positive rate, and ROC curves.

We included recall in the revision. All true-positives were found across all methods. Like a real-world experiment, only few (10) SNPs are causal; and in absolute number, the number of false positives typically ranged from 0 to 50 (excluding number of expected false positives); PCA deviated more and had up to 300 for TGP and sparse settings.  Rarely did a method not capture all true positives, and if so, it only missed one.

Regarding false positive rate, it gives a similar signal as the measured precision. This is because as mentioned, the number of true positives was roughly the same across methods. Because precision is the number of detected true positives over the number of detected true positives and false positives, precision simply differed by a method's number of false positives. We also did control for the number of expected false positives; this is clarified in Appendix D's revision.

> R3: * How do the authors call a variant to be associated with the phenotype (y)? More specifically, what is the distribution of the null hypothesis?

We revised Appendix D. In summary, we followed Song Hao Storey (2015), which calculates a likelihood ratio test statistic for each SNP. It is the difference of the maximum likelihood solution on the trait model to the maximum likelihood solution on the trait model with influence of the SNP fixed to zero. This null has a chi^2 distribution with degrees of freedom equal to the number of weights fixed at zero (equal to the number of hidden units in the first layer).

---

### Decision · Program_Chairs · 2018-01-29
**ICLR 2018 Conference Acceptance Decision**

**Decision:**

Accept (Poster)

**Comment:**

The reviewers agree that the work is high quality, clear, original, and could be significant.

Despite this, the scores are borderline. The reason is due to rough agreement that the empirical evaluations are not quite there yet. In particular, two reviewers agree that, in the synthetic experiments, the method is evaluated on data that is an order of magnitude too easy and quite far from the nature of real data, which has a much lower signal to noise ratio.

However, the authors have addressed the majority of the concerns and there is little doubt that the authors are capable of carrying out this new experiment and reporting its results. Even if the results are surprising, they should shed light on what seems to be an interesting new approach.